# UV Radiation in DNA Damage and Repair Involving DNA-Photolyases and Cryptochromes

**DOI:** 10.3390/biomedicines9111564

**Published:** 2021-10-28

**Authors:** Yuliya L. Vechtomova, Taisiya A. Telegina, Andrey A. Buglak, Mikhail S. Kritsky

**Affiliations:** 1Bach Institute of Biochemistry, Research Center of Biotechnology of the Russian Academy of Sciences, 119071 Moscow, Russia; telegina@inbi.ras.ru (T.A.T.); mkritsky@inbi.ras.ru (M.S.K.); 2Faculty of Physics, Saint Petersburg State University, 199034 Saint Petersburg, Russia; andreybuglak@gmail.com

**Keywords:** DNA repair, cancer, ultraviolet, ROS, DNA-photolyase, cryptochrome, molecular evolution

## Abstract

Prolonged exposure to ultraviolet radiation on human skin can lead to mutations in DNA, photoaging, suppression of the immune system, and other damage up to skin cancer (melanoma, basal cell, and squamous cell carcinoma). We reviewed the state of knowledge of the damaging action of UVB and UVA on DNA, and also the mechanisms of DNA repair with the participation of the DNA-photolyase enzyme or of the nucleotide excision repair (NER) system. In the course of evolution, most mammals lost the possibility of DNA photoreparation due to the disappearance of DNA photolyase genes, but they retained closely related cryptochromes that regulate the transcription of the NER system enzymes. We analyze the published relationships between DNA photolyases/cryptochromes and carcinogenesis, as well as their possible role in the prevention and treatment of diseases caused by UV radiation.

## 1. Introduction

The constant destructive impact of various adverse environmental factors (pollution with toxic substances, various types of natural and artificial radiation, etc.) causes the disturbance of the normal functioning of living cells [1,2,3]. This leads to the development of various diseases, which can eventually result in chronic ones. In turn, the immune and endocrine systems cease to cope with their protective and regulatory functions due to an increase in constant load, and, ultimately, all this can lead to more serious disorders, including cancer. A necessary condition for the occurrence and development of the process of carcinogenesis is DNA mutations. DNA mutations can occur due to mutagenic environmental factors (in particular, UV) when the effective repair does not function [4,5,6]. A person is exposed to intense UV light both in connection with professional activities that require a long stay out of doors and as a result of following fashion trends, sunbathing on the beach, or using special lamps in tanning salons. Prolonged exposure to UV on the skin leads to hyperpigmentation, photoaging due to collagen fiber damage [7], and the accumulation of mutations in the cell’s DNA. About 90% of non-melanoma skin cancers and 86% of melanomas are associated with chronic UV irradiation of the skin [8]. UV inhibits the synthesis of ATP and disrupts the immune response, which also contributes to carcinogenesis [8,9]. It was found that chronic exposure to solar radiation is the most important environmental factor involved in the pathogenesis of actinic keratosis and squamous cell carcinoma [9]. It has also been shown that chronic UV radiation in farmers is associated with a high risk of the earlier development of basal cell carcinoma and its aggressive subtypes [10]. Therefore, consideration of issues related to the mutagenic effect of UV radiation on DNA, mechanisms of DNA repair, as well as consideration of approaches preventing pathological effects of UV radiation is relevant.

## 2. Possible Mechanisms of DNA Damage by UV Radiation

### 2.1. Target Molecules for UV Exposure

Targets of UV radiation in living organisms can be various photoactive molecules, for example, pterins, folates, flavins, porphyrins, aromatic amino acids, etc. (Figure 1), as well as biopolymers: proteins and nucleic acids [11,12]. Such photoactive molecules transfer into an excited state after the light absorption, and the excess energy can be utilized in several ways, among which three main options are of biological significance.

(1) If the molecule is a chromophore of a photoreceptor protein, then the absorbed energy is converted into a signal to trigger various processes. For example, flavin is a chromophore of cryptochromes (CRYs), which are involved in the photoregulation of circadian rhythms (see Section 6).

(2) Important biologically active molecules can undergo partial chemical modification (as happens with DNA, see Section 2.2 and Figure 2) or be completely destroyed, which can lead to a deficiency of these molecules in the body. This is especially true for those substances that cannot be synthesized in the human body. For example, folic acid derivatives are destroyed by UV radiation. This may be the reason for the deficiency of this vitamin in people with fair skin, exposed to increased impact to solar radiation [13].

(3) Light-excited molecules can become sensitizers for the destruction of other molecules or lead to the formation of reactive oxygen species (ROS). ROS, in turn, can lead to damage to other molecules, including DNA (see Section 2.3 and Figure 2) and lipids [14]. Porphyrins (Figure 1) are well-known photosensitizers in medicine, and their ability to generate ROS underlies the photodynamic cancer therapy [15].

### 2.2. Damaging Effect of UVB Light on DNA

The most dangerous impact of UVB (280–320 nm) radiation is the DNA damage in skin cells. The main products of UVB-irradiated DNA are cyclobutane pyrimidine dimers (CPD, 75%) and pyrimidine-pyrimidone (6-4) photoproducts ((6-4)PP, 25%), in which two neighboring pyrimidines are covalently bound (Figure 2). In addition, irradiation can lead to breaks in one or both of DNA chains at once. The formation and accumulation of CPD and (6-4) PP block the DNA replication and transcription, which disrupts the normal functioning of cells. If these lesions are not removed in a timely manner, this can lead to cell death and the occurrence of inflammatory processes at the tissue level. In some cases, errors in the repair process can lead to the appearance and accumulation of mutations (see Section 3), which can subsequently cause various diseases and skin cancer [16,17,18,19,20].

### 2.3. The Influence of UVA Light on the Processes Occurring in Skin Cells

It is believed that UVA radiation (320–400 nm) is not as harmful as UVB. In small doses, UVA light is necessary for a human because it is a signal for various photoregulatory proteins, in particular for restarting circadian rhythms, which, in turn, regulate many different processes in the body (see Section 6 and Figure 2). However, in high doses, UVA radiation can lead to the suppression of the immune system and the formation of ROS through photosensitization reactions [9,14,18,21]. ROS cause oxidative stress and photoaging, and this can also lead to skin cancer [20]. DNA can be damaged by ROS to form 8-oxo-7,8-dihydro-2’-deoxyguanosine (8-OHdG) (Figure 2), which, such as CPD, interferes with normal cell functioning [9,14,18,20,21]. ROS activate the expression of matrix metalloproteinases, which cause degradation of collagen fibers, leading to the appearance of wrinkles and skin aging [7,20,22,23].

A special pathway of the destructive effect of UVA radiation on DNA is caused by cell damage in the presence of melanin (Figure 2). Back in 2003, it was shown that oxidative damage is not the main type of UVA-induced damage in skin cancer. Thus, oxidized pyrimidines, single-chain breaks, oxidized purines (8-OHdG), and CPD are formed in a ratio of 1:1:3:10. Moreover, it was found that UVA generates CPD with a large predominance of thymine dimers, which indicates their formation through photosensitized triplet energy transfer [24]. In further studies, it was shown that melanin is the photosensitizer of the process. Melanin properties are two-fold: 1) when the melanin synthesis is completed, melanin in a certain geometric configuration in keratinocytes performs a protective function, but 2) it can have pro-oxidant properties during partial polymerization in melanocytes when exposed to UV radiation. The potential role of UVA in skin carcinogenesis is also confirmed by epidemiological studies showing an increased risk of melanoma among the users of tanning lamps producing UVA radiation [25].

In pigmented melanocytes, CPD occurs both instantly and within a few hours after UV irradiation in the dark. The path of CPD occurrence in the dark is partially similar to bioluminescence and is as follows. UV activates nitric oxide synthases, which generate a nitric oxide radical (NO^•^), and NADPH oxidases, which generate superoxide-anion radical (O_2_^−^). Further, these radicals interact with each other to form strong oxidant peroxynitrite (ONOO^−^) (Figure 2). Peroxynitrite oxidizes melanin while exciting the melanin electron to a high-energy level. This is the process of “chemical electron excitation” in melanin: a non-radiative triplet energy transfer to DNA occurs with the formation of CPD [26,27]. UVA and peroxynitrite contribute to the solubilization of melanin and increase the permeability of the nuclear membrane to melanin. This pathway can be considered as a melanin-dependent pathogenesis of melanoma. In the same way, the chemical excitation of melanin can trigger pathogenesis in other tissues, in which nitric oxide and superoxide anion radicals arise in cells containing melanin [28]. It is important to note that photodynamic therapy of skin cancer with red light does not cause CPD formation in the presence of melanin [29].

Exposure to strong UV radiation is the main etiological environmental factor for all forms of skin cancer, including melanoma. The ability to repair DNA determines the risk of skin cancer. The sensitivity of cells to the severe effects of UV radiation depends on the degree of skin pigmentation. In turn, the process of melanogenesis can be disrupted when exposed to various exogenous etiological environmental factors, including UV [30]. Disruption of melanogenesis occurs in a number of dermatological diseases, including vitiligo [31,32,33]. The study of melanogenesis and the ways of its regulation are important for the development of new photoprotective strategies for the prevention of skin cancer.

## 3. DNA Repair Systems Involved in the Photodamage Removal

Cells have many different mechanisms for repairing each type of DNA damage that occurs spontaneously and is caused by exogenous factors [6,34]. The main mechanisms of DNA repair include: direct repair, when the enzyme restores the original structure without removing damaged nucleotides; excision repair, through the removal of damaged sites, followed by the synthesis of new nucleotides: base excision repair (BER) and nucleotide excision repair (NER); repair of unpaired bases (mismatch repair); repair of single-strand and double-strand breaks [6,34]. To remove photodamage, both direct repair with the participation of the DNA photolyase [4,35,36] and NER for CPD and (6-4)PP or BER for 8-OHdG [4,20,37] are used (Figure 3).

In humans, as in most mammals (except for some marsupials), there are no DNA photolyases and the only system responsible for removing the most dangerous CPDs for DNA is NER. It is known that defects in the NER process lead to a xeroderma pigmentosum disease (XP). People suffering from this disease are extremely sensitive to sunlight and are susceptible to the development of skin oncological diseases. Genetic analysis of such patients revealed the mutations in seven main genes, called XPA-XPG, responsible for the NER function [35]. In general, more than 20 different proteins take part in the NER process, and it can be triggered in two ways. The first way is global genome NER when the NER enzymes (in humans, these are complexes of RPA, XPA, and XPC proteins) themselves find damaged sites and start the repair process. The second way is transcription-coupled NER, where, during transcription, an RNA polymerase bumps into a damaged site and initiates the repair process. Next, a cascade of reactions involving a protein complex is started, as a result of which endonucleases (XPF and XPG) cut out a DNA oligomer of about 30 nucleotides containing CPD. Then, a polymerase is attached and synthesizes a complementary intact DNA chain. The ligase completes the process by connecting the free ends of DNA chains (Figure 3) [4,35,37,38,39]. Both pathways require the involvement of a large number of enzymes, and as a result, the process stretches over time to several hours, thus the processes triggered by CPD dimers, for example, melanogenesis, have time to start in cells. In addition, the repair process itself often occurs with violations and leads to mutations. In such erroneous CPD repair, mutations with cytosine-to-thymine replacement occur most often, which is characteristic for mutations found in cancer cells [9,17,40,41].

The BER system is used to remove 8-OHdG formed as a result of photo-oxidative stress (Figure 3). The BER scheme is similar to the NER scheme but includes fewer enzymes. The damaged base is removed by 8-oxoguanine glycosylase, then from 1 to 13 nucleotides near the damaged site are removed by AP endonuclease and synthesized again. Disruption in the BER function leads to fetal mortality or predisposition to cancer and neurological symptoms in animals. NER is able to partially replace BER in case of violations in its operation and also stimulate the enzymatic activity of some BER factors [37].

Furthermore, sometimes the NER system cannot recognize CPD and restore DNA, in this case, a rough “SOS” repair system works: during replication, the affected DNA sections are bypassed and subsequently replaced with nucleotides that are not complementary to the original chain, which can also lead to mutations [42]. Most of the spontaneously occurring mutations that accumulate in cells throughout a person’s life can go unnoticed without any serious consequences, but some of them can change key cellular functions and lead to cancer and aging [9].

Understanding the function of repair systems is important not only because of possible disruptions in their work that lead to the occurrence of oncological diseases but also because of possible mechanisms and places of application in the treatment of these diseases. Many cancer treatment strategies are aimed to destroy the DNA of tumor cells, and in this case, the repair systems existing in these cells will reduce the effectiveness of such treatment [6]. At the same time, during chemotherapy and radiotherapy, not only diseased cells are often affected, but also healthy ones. The ability to speed up the process of restoring these cells after the end of treatment (during the rehabilitation period) will also improve the quality of treatment.

## 4. Proteins of the DNA-Photolyase/Cryptochrome Family (CPF)

### 4.1. Structure and Functions of CPF

DNA photolyases are enzymes that repair DNA damaged by UVB light (280–320 nm). DNA photolyases operating under the action of near-UV and blue light (320–480 nm) are part of the DNA photolyase/cryptochrome family (CPF). Cryptochromes (CRYs) are protein receptors of near-UV and blue light. Despite the phylogenetic relationship, CRYs perform completely different functions from DNA photolyases. CRYs photo-regulate the transcription of various genes. In addition, CRYs are a part of the central circadian oscillator of animals and participate in the regulation of circadian rhythms by light in both plants and animals (including humans) [35,43,44,45,46,47]. In addition, CRYs participate in magnetoreception [48,49].

The proteins of the CPF family have a similar structure and contain two non-covalently bound chromophores in a stoichiometric proportion. These are monomeric globular proteins consisting of two domains: N-terminal α/β domain and C-terminal α-helical domain. The C-terminal α-helical domain of all members of the family contains: (1) a conservative sequence of amino acids; (2) a binding site of the flavin chromophore; (3) the binding site of the substrate (DNA photoproducts in DNA photolyases and ATP in CRY) [50,51,52,53]. The main chromophore, flavin adenine dinucleotide (FAD), is located in the active center of all CPF proteins. It is responsible for binding to the substrate and the main photoreceptor function. The second chromophore (5,10-methenyl-5,6,7,8-tetrahydrofolate, 7-desmethyl-8-hydroxy-5-dezazariboflavin, 6,7-dimethyl-8-ribityllumazine, FMN or the second FAD molecule, depending on the organism), which performs the function of a “light harvesting antenna” that captures additional light and transfers excitation energy to FAD, is located between the two domains and is not found in all representatives of this family [50,51,54].

The amino acid sequence of CRYs is 25–40% homologous to the DNA-photolyase sequence, while in most cases, the active site has lost the ability to bind DNA (this is typical for plant and animal CRYs), although some groups of CRYs (such as DASH, etc.) have retained fully or partially the functions of DNA repair. [47,51,52]. There is an additional “tail” at the C-end of CRY, which is essential for protein localization and interaction with other regulatory proteins, by binding to which CRY can influence the gene expression process [46,55,56,57].

In many organisms, DNA photolyases are located in tissues that are not directly exposed to light. Nevertheless, they are present there and perform some functions. It is assumed that there are two possible functions in this case. The first is that excision DNA repair enzymes that work in the dark sometimes do not recognize CPD due to minor changes in the DNA structure, especially in areas of tightly packed DNA covered with chromatin [39]. However, such a system is able to recognize a DNA photolyase that binds to such a DNA site and does not have the ability to repair DNA due to the lack of light. After that, it only remains to remove this DNA site together with the photolyase and restore the original structure. The second function is that photolyases can bind some drug-damaged DNA sites, for example, cisplatin-dGpG diadduct, which is not recognized by excision DNA repair enzymes. Thus, even in the absence of light, DNA photolyases can promote cell survival and eliminate DNA damage [45].

### 4.2. The Reaction Mechanism of DNA-Photolyase

The reaction mechanism of DNA-photolyase is classical enzymatic catalysis in which one of its stages depends on light [35,45]. At the first stage, the enzyme binds to the damaged DNA and embeds them into the active center of the enzyme to form a stable enzyme-substrate complex. This stage is independent of light. The binding site itself is a pocket on the surface of the α-helical domain. At the bottom of this pocket, there is FAD in an unusual U-shaped conformation. Such a structure is complementary to the structure of the damaged site and makes it possible to firmly bind CPD (the binding constant is equal to 10^−9^ M), forcing the damaged sites to “turn out” from the structure of the DNA double helix [17,45,54]. Such a strong interaction of DNA photolyases with the substrate is used for analytical and diagnostic purposes when fluorescently labeled photolyases are CPD markers in the studied samples or living cells [58].

The second stage occurs when the antenna molecule absorbs a photon of UVA/blue light and transfers the excitation energy (by the Förster dipole-dipole resonance interaction) to the FADH^-^, which then transfers the electron to the CPD or (6-4)PP and thus repairs the DNA. After that, the electron returns to the flavin, regenerating the FADH^-^ form. The entire photocatalysis process occurs in 1.2 ns [35,59]. At the last stage, the restored DNA, which no longer has sufficient affinity for the photolyase binding site, dissociates from the enzyme [54].

Unlike CPD photolyases, in (6-4) photolyases, the incorporation of the substrate into the active enzyme complex apparently turns the (6-4)PP into an oxytane cycle [60]. Further splitting of this cycle is more complicated than splitting CPD, and, therefore, the quantum yield of photoreparation of (6-4) photolyase is less (0.3) than that of CPD photolyase (0.7–1). Otherwise, the reaction mechanisms of (6-4) and CPD photolyases coincide [45,60,61].

## 5. DNA Photolyases as a Tool of Protection against Photodamage of Human Skin and Its Treatment

The effectiveness of using sunscreens in preventing skin cancer is well known. However, compliance with the rules of regular sunscreen usage is a problem, mainly due to the poor cosmetic qualities and the cost. In addition, there are concerns about the possible harmful effects of some sunscreen components on the person oneself and the environment. Recent developments in the field of sunscreens creations have led to significant improvements in the texture, photostability, water resistance, and effectiveness of sunscreens. An increasing number of sunscreens contain antioxidants, herbal extracts, lichens, and various biomolecules as photo-protection enhancers. Thus, it has been shown that sunscreens containing vitamin D and B_3_, as well as E as an antioxidant, improve the condition of the skin after irradiation and contribute to the repair of damaged DNA [8,62]. To improve the effectiveness of sunscreens, enzymes that stimulate the NER (T4 endonuclease V) and BER (8-oxoguanine glycosylase) are added to the sunscreens. T4 endonuclease V is an enzyme that has been isolated from *Escherichia coli* infected with the bacteriophage T4. It initiates DNA repair at the site of UV-induced CPD, amplifying it by four times. In addition, the enzyme stimulates skin regeneration and prevents the destruction of extracellular matrix components, which helps prevent photoaging [20].

Humans do not have their own DNA photolyase, thus it was proposed to use an extract containing DNA photolyase from *Anacystis nidulans* to expand the photoprotective functions of sunscreens [9,16,63,64,65]. To deliver the enzyme through the stratum corneum of human skin and introduce photolyase into the living epidermis, it is encapsulated in liposomes. It has been shown that the use of photolyase rapidly reduces the amount of CPD in skin cells and supports intracellular regeneration. Intracellular regeneration inhibits apoptosis caused by UV radiation and reduces skin inflammation caused by sunlight exposure through the inhibition of the pro-inflammatory cytokine interleukin 6 [20]. Films containing DNA photolyases are also considered as a treatment for UV-induced human skin diseases, including oncological ones [66]. At the moment, no facts have been revealed that photolyase can negatively affect a person in any way. However, extracts isolated from *A. nidulans* often contain lipopolysaccharides of this bacterium in addition to the photolyase itself, which can cause allergic reactions, respiratory, or skin diseases in humans [9,67]. In recent years, it has been proposed to use photolyases isolated from extremophiles found in the Arctic as an alternative to the extract of photolyase from *A. nidulans*. Recombinant photolyase purified by affinity chromatography with immobilized metal has no impurities and shows high reactivity and resistance to high doses of UV [67]. Different variants of DNA photolyase immobilization on various nanostructured carriers for targeted delivery and stability of the molecule are also considered [68]. In addition, for example, palladium/platinum nanoparticles themselves can be used to eliminate the effects of photodamage and oxidative stress in cells [69,70,71].

Irradiation of the skin with UV light and the accumulation of CPD trigger the process of melanogenesis; as a result, melanin is synthesized, and a tan is formed. Melanin is able to partially absorb UV light, thus performing protective functions. However, as previously described, melanin can also be carcinogenic, contributing to the formation of CPD even after irradiation termination [8,26]. If you use sunscreens with photolyase, then the photorepair occurs so quickly (almost immediately after application) that the signal for the production of melanin may not have time to pass, and the person may not tan. If the stimulation of NER enzymes is used in sunscreens, then the repair is still quite slow (about 6 h), and the person has time to tan [72]. In addition, the efficiency of creams containing T4 endonuclease V in removing CPD is much lower (about 20%) than of those containing photolyase (more than 40%) [9]. Therefore, the rapid removal of CPD and prevention of sunburn (melanin production) with the help of DNA photolyases may also be a strategy protecting people prone to skin cancer.

Modern dermatology, skin biology, and methods of preventing skin aging suggest that a good sunscreen should: (1) protect against both UV-B and UV-A radiation; (2) contain stable and safe filters that do not harm the environment; (3) contain antioxidants for ROS removal; and (4) include cellular DNA repair enzymes. A sunscreen containing a complex of DNA repair enzymes (photolyase, endonuclease, and 8-oxoguanine glycosylase) and a powerful antioxidant complex (carnosine, arazine, ergothionine) has shown high effectiveness against the damaging effects of UV radiation and preventing photoaging of the skin and the occurrence of non-melanoma skin cancer [20,73]. It is also worth noting that creams containing DNA photolyase are recommended to be used not only to prevent the harmful effects of UV radiation but also as a treatment for already existing skin lesions, such as actinic keratosis [9,64].

The use of DNA photolyases (missing in humans) for the treatment of diseases caused by hereditary or acquired disorders in the NER function is considered from the point of view of gene therapy. Thus, it was shown that mice with an artificially introduced and expressed DNA photolyase gene show greater resistance to the damaging effects of UV radiation and are less prone to cancer [38]. It was also demonstrated in [41] that pseudouridine-containing mRNA encoding *Potourus* photolyase synthesized in vitro is a powerful tool for the rapid repair of DNA photodamage in human keratinocytes. These data suggest that gene therapy methods can be used in the future to treat diseases caused by hereditary or acquired disorders in the DNA photodamage repair system.

## 6. Cryptochromes, Circadian Rhythms, and the NER Regulation

Cryptochromes are one of the components of the circadian clock of animals and plants. The essence of the mechanism of circadian rhythms is that the PER (period) and CRY circadian rhythm proteins when accumulated in the cytoplasm in a certain amount, are combined into a complex that inhibits their own biosynthesis, as well as the biosynthesis of other regulatory proteins (Figure 4). Over time, the number of these proteins becomes smaller, and biosynthesis starts again. Thus, an approximately 24 h cycle ends [35]. Circadian rhythms regulate key aspects of cell growth and survival, responses to genotoxic stress, cellular aging, and metabolism. It is known that disruption of circadian rhythms, for example, night work, increases the risk of cancer and also contributes to the progression of tumors. On the contrary, the activity of some genes and factors associated with circadian rhythms is disrupted in cancer patients. In this connection, such treatment regimens are proposed when the use of antitumor drugs is combined with drugs that regulate circadian rhythms or occur at a certain time of the daily cycle [74,75].

In animals, there is a central oscillator, which is located in the brain and is responsible for the rhythmic behavior of the body as a whole and the synchronization of the oscillators of individual organs. The receptor responsible for restarting the cycle under the influence of light is located in the eyes [46,56,75,76]. CRYs of some insects’ function as photoreceptors, participating in the process of circadian rhythm restart. In mammals, the most likely candidate for the role of a photoreceptor restarting the central oscillator is the photopigment melanopsin, which is contained in small amounts in retinal ganglion cells [75,77]. In addition to direct participation in the formation of the circadian cycle, the proteins of this cycle interact with other proteins that regulate the vital functions of the cell. Thus, CRY participates in the regulation of glucose levels by interacting with glucocorticoid receptors [75,78].

It has been shown that the expression of one of the key human NER enzymes, XPA, is regulated by circadian rhythm proteins [35]. CRY1 and CRY2 play different roles in responses to genotoxic stress: cells lacking CRY1 showed an increased response, while cells without CRY2 showed a reduced response and accumulated the damaged DNA [35,79]. The inhibition of the response to genotoxic stress by CRY1 is explained by the fact that it competes with the ATR-CHK1 protein (DNA damage checkpoint factors) for binding to the TIMELESS (TIM) cell cycle protein. These TIM and ATR-CHK1 proteins, in turn, form a complex that participates in the transmission of a DNA damage signal to trigger the transcription of the NER proteins (in particular, XPA) [76,80]. These results suggest that although CPF proteins have lost the DNA photorepair functions in mammals, cryptochromes are still involved in DNA repair regulating the transcription of the NER proteins.

## 7. Evolutionary Aspects of UV-Damaged DNA Repair Pathways

Since it is impossible to fully understand the essence of biological processes and phenomena without evolutionary consideration, we should briefly touch upon the evolution of the CPF proteins. CPF proteins are found in all three kingdoms of living organisms (bacteria, archaea, and eukaryotes). They have a similar structure and a common origin, probably from a common ancestor, which could have both the functions of DNA repair and regulation of gene transcription [47]. The formation of such a common ancestor probably occurred on Earth about 3.5 billion years ago under intense UV radiation in a reducing atmosphere [81]. In such conditions, the organisms required a rapid DNA repair system to protect them from damaging UV radiation. Later, with the formation of the ozone screen, the urgent need for rapid repair with the help of DNA photolyases disappeared, and many organisms, under the pressure of certain evolutionary factors, completely or partially lost the photorepair ability. For example, the yeast of the *Saccharomycotina* subtype, which followed the path of rapid evolution due to mutations and a decrease in the genome size, completely lost not only the photolyase genes but also the genes of other DNA repair systems [82].

The NER repair system was formed after the separation of three kingdoms of living organisms: the proteins involved in the work of this system have completely different origins in all three kingdoms. The representatives of these three kingdoms have multiple differences in the repair mechanism. However, unlike photolyases, which are designed to remove only a limited type of damage, the NER system is able to remove a wide range of DNA damage, replacing not only photolyases but also BER if necessary [34,37]. The emergence of NER is believed to be the result of the response of living organisms to the complication of the biogenic chemosphere [34]. Multiple damaging factors appeared in the biogenic chemosphere, and they had to be removed by a more universal system, rather than inventing a separate removal system for each damage. At the same time, such versatility and multitasking greatly complicated the process of DNA repair and reduced its speed. This, in turn, affects the quality of CPD removal in humans when exposed to large doses of UV radiation and often leads to skin diseases and its photoaging.

The absence of DNA photolyases in mammals may indicate that natural selection has become weaker at this stage of evolution. The absence of mutations in animals with a relatively small number of offspring and the rate of their maturation could have a negative impact on the adaptation of organisms to the changing living conditions [56]. UV protection with the help of photolyases, narrowly focused on the repair of only one type of DNA damage, is not so necessary for animals with pelage or protective skin pigmentation. At the same time, other tasks such as the regulation of circadian rhythms and the expression of genes, including the NER protein genes, continue to be relevant. Therefore, the evolution of CPF proteins has gone towards fixing the functions of CRYs in mammals. Although, as a result of long-term evolution in humans, cryptochromes have mostly lost the functions of a photoreceptor, and melanopsin in the retina has become the main physiological photoregulator, the role of CRY as stabilizer and regulator of the signal phototransduction cascade has remained [77].

## 8. Conclusions

In the actual conditions of an increase in the influence of UV light and the deteriorating environment, it is not clear that the use of photolyases preventing photoaging of the skin could avoid or treat skin diseases caused by UV damage to DNA. Sunscreens containing photolyase have shown high effectiveness, but they have not been widely used, apparently due to the relatively high cost and lack of stability of the protein extracts. Improving the properties of such creams and a wider distribution should increase their preventive role. It is possible that the use of genetic engineering methods and the creation of photolyase mRNA can help people suffering from severe forms of diseases caused by the disruption of the NER system, such as xeroderma pigmentosum. Humanity is actively preparing to explore outer space, where the influence of various types of radiation on DNA will be intense. It should be important to take a closer look at the possibility of rapid DNA photorepair with DNA photolyases. In addition, the accurate study of the influence of circadian rhythms and the role of cryptochromes in the regulation of cell functions, prevention, and treatment of diseases caused by UV radiation, should be important.

## Figures and Tables

**Figure 1 biomedicines-09-01564-f001:**
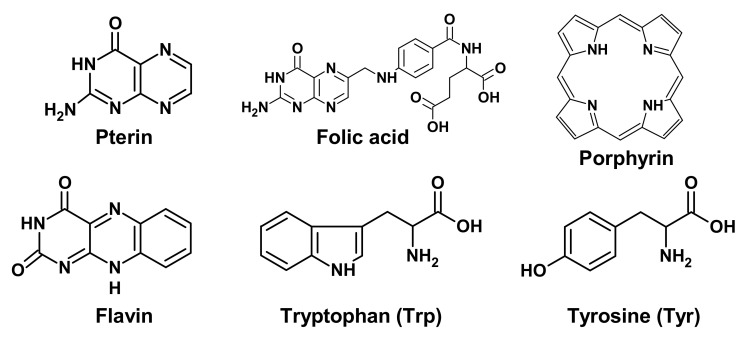
Chemical formulas of low molecular weight biological chromophores.

**Figure 2 biomedicines-09-01564-f002:**
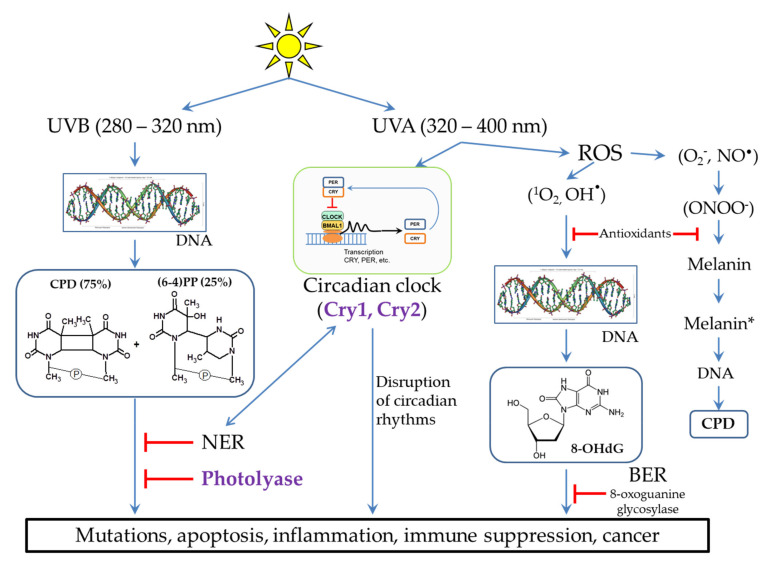
Different UV radiation of DNA leads to different lesions and subsequent occurrence of pathological situations. Methods of lesion prevention and repair are presented. * - The excited state of melanin.

**Figure 3 biomedicines-09-01564-f003:**
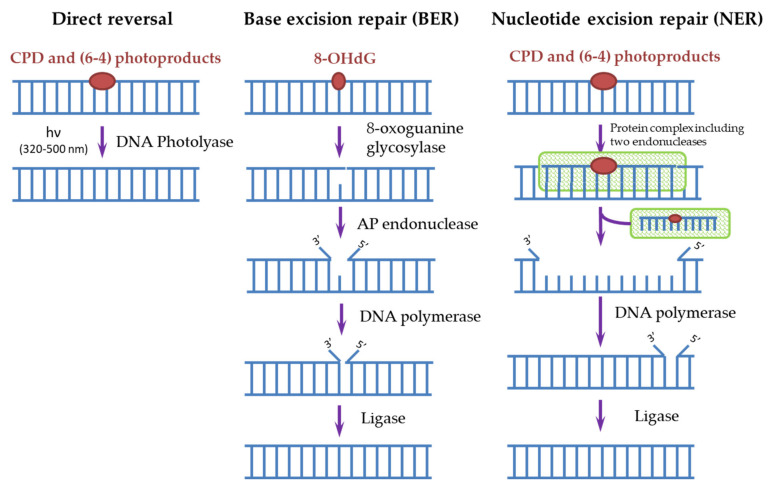
Scheme of direct and excisional repair in photodamaged DNA.

**Figure 4 biomedicines-09-01564-f004:**
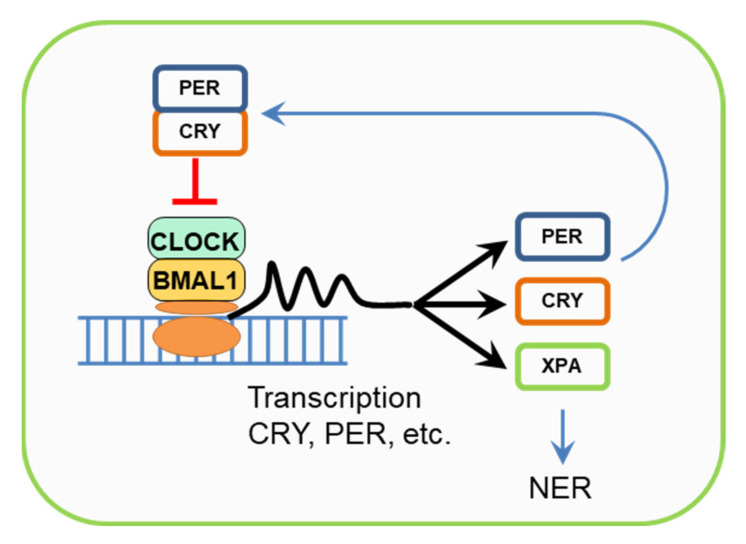
Mechanism of the mammalian circadian clock. CLOCK and BMAL1 are transcription activators of Cry and Per genes. CRY and PER are transcriptional repressors. They inhibit their own transcription, thus causing the rise and fall of CRY and PER levels with 24-h periodicity.

## Data Availability

The review does not use new unpublished experimental data.

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
