# Peer review of "UV Radiation in DNA Damage and Repair Involving DNA-Photolyases and Cryptochromes"

_biomedicines, 2021, doi:10.3390/biomedicines9111564_

Round 1
Reviewer 1 Report
The article addresses an important and current issue: that of the cause-effect relationship between UV radiation and neoplastic and pre-neoplastic skin lesions and the possibility, based on the known mechanisms of action, to fight this relationship.
However, it needs review to simplify and organize objectives and conclusions:
1. the title is too generalist.
2. The abstract seems more like an introduction, does not speak of cutaneous tumors (in which the whole article focuses) and is independent of the conclusions.
3. The description of the mechanisms is too extensive.
4. Line 62: the paper reference 17 does not explain why it considers related basal cell carcinoma, associated with acute sun exposure during childhood and adolescence. Most articles consider basocellular associated with chronic and repeated exposure to the sun. This statement would need a better explanation.
5. Line 79. An article doesn't need colloquial expressions. It shouldn't be a question-answer succession.
6. Line 133-136: "In when" does not strike me as an appropriate expression for a definition.
7. Line 328: Is it really necessary to include commercial names?
8. Line 423 – It would be preferable, rather than "prevent and treat diseases caused by UV damage to DNA" use "preventing photo-aging of the skin and the occurrence of non-melanoma skin cancer caused by UV damage to DNA"
Author Response
Replies to Reviewer 1 Comments.
The authors are grateful to the reviewer for a careful analysis of the manuscript: biomedicines-1383821 “UV radiation in DNA damage and repair”. We have corrected the manuscript in accordance with the obtained comments. The manuscript has been corrected by an English language professional. Changes in the text were marked using the “Track changes” function of MS Word and commented below.
|
Referee’s comment |
Answer |
|
The article addresses an important and current issue: that of the cause-effect relationship between UV radiation and neoplastic and pre-neoplastic skin lesions and the possibility, based on the known mechanisms of action, to fight this relationship. However, it needs review to simplify and organize objectives and conclusions. |
We sincerely thank the reviewer for a thorough analysis of our work and for valuable comments that allowed us to change our review for the better. |
|
The title is too generalist. |
We changed the title to “UV radiation in DNA damage and repair involving DNA-photolyases and cryptochromes” |
|
The abstract seems more like an introduction, does not speak of cutaneous tumors (in which the whole article focuses) and is independent of the conclusions. |
This has been corrected in the revised manuscript. |
|
The description of the mechanisms is too extensive. |
We have reduced some details that are not essential for this review in the reaction mechanism of DNA-photolyase |
|
Line 62: the paper reference 17 does not explain why it considers related basal cell carcinoma, associated with acute sun exposure during childhood and adolescence. Most articles consider basocellular associated with chronic and repeated exposure to the sun. This statement would need a better explanation. |
Thank you for this comment. We agree with this remark, so this statement has been removed from the text. |
|
Line 79. An article doesn't need colloquial expressions. It shouldn't be a question-answer succession. |
We agree with this comment, so this question has been removed from the text. |
|
Line 133-136: "In when" does not strike me as an appropriate expression for a definition. |
The phrase has been changed to: “The second way is transcription-coupled NER where, during transcription, an RNA polymerase bumps into a damaged site and initiates the repair process.” |
|
Line 328: Is it really necessary to include commercial names? |
We agree with this comment, so commercial names have been removed from the text. |
|
Line 423 – It would be preferable, rather than "prevent and treat diseases caused by UV damage to DNA" use "preventing photo-aging of the skin and the occurrence of non-melanoma skin cancer caused by UV damage to DNA" |
This has been corrected in the revised manuscript. |
Reviewer 2 Report
Revision to manuscript “UV radiation in DNA damage and repair” by Yuliya L. Vechtomova et al.
Comments and suggestions for authors:
- The title is too generic, it should be focused on cryptochromes and photolyases in relation to DNA damage, which is the main theme of this review.
- The connection between the hole in the ozone layer and DNA damage is unnecessary, and aside the introduction is not mentioned anymore. I suggest focusing the introduction more on skin pathologies related to photoaging and UV damage, which are well explained within the whole text.
- In the introduction section psychological stress is erroneously indicated as adverse environmental factor.
- In the introduction section the link between the list of chromophores and skin cancer is not clear as well as the link with the main products of UVB irradiated DNA. I suggest to move this list and the relative figure in section 4 (Proteins of the DNA photolyase/cryptochrome family).
- In general the introduction section should be re-organized. The title refers to the mechanism of DNA damage induced by UV radiations, however those mechanisms are not well explained, some paragraphs are not well connected, the list of chromophores in this section is irrelevant and the reaction using melanin as photosensitizer is too detailed as compared to other mechanisms described. However, Figure 2 describes well the idea. I suggest to better describe the figure to improve this section.
- I suggest the following running title for the section 3: “DNA repair systems involved in the removal of photodamage”
- Probably some typos (cyrillic fonts?) are reported in the figure 3
- Section 4 is too long, is difficult for the reader to assimilate all the information. I suggest to split it into two sections: one relative to the members of the protein family and one relative to the reaction mechanisms
- Section 7 on my opinion is completely unrelated to the notions explained before. If the evolution concept is considered relevant it could be summarized in few lines and added in the introduction section.
Author Response
Please see the attach
Replies to Reviewer 2 Comments.
The authors are grateful to the reviewer for a careful analysis of the manuscript id: biomedicines-1383821 “UV radiation in DNA damage and repair”. We have corrected the manuscript in accordance with the obtained comments. The manuscript has been corrected by an English language professional. Changes in the text were marked using the “Track changes” function of MS Word and commented below.
|
Referee’s comment |
Answer |
|
The title is too generic, it should be focused on cryptochromes and photolyases in relation to DNA damage, which is the main theme of this review. |
We changed the title to “UV radiation in DNA damage and repair involving DNA-photolyases and cryptochromes” |
|
The connection between the hole in the ozone layer and DNA damage is unnecessary, and aside the introduction is not mentioned anymore. I suggest focusing the introduction more on skin pathologies related to photoaging and UV damage, which are well explained within the whole text. |
We agree with this comment and the introduction has been changed. |
|
In the introduction section psychological stress is erroneously indicated as adverse environmental factor. |
This has been corrected in the revised manuscript. |
|
In the introduction section the link between the list of chromophores and skin cancer is not clear as well as the link with the main products of UVB irradiated DNA. I suggest to move this list and the relative figure in section 4 (Proteins of the DNA photolyase/cryptochrome family). |
Thank you for this comment. This comment seems to be talking not about the introduction, but about section 2 (Possible mechanisms of DNA damage by UV radiation). We have expanded and reorganized this section to make the relationship between biological chromophores, main products of UVB irradiated DNA and the occurrence of various diseases more understandable. |
|
In general the introduction section should be re-organized. The title refers to the mechanism of DNA damage induced by UV radiations, however those mechanisms are not well explained, some paragraphs are not well connected, the list of chromophores in this section is irrelevant and the reaction using melanin as photosensitizer is too detailed as compared to other mechanisms described. However, Figure 2 describes well the idea. I suggest to better describe the figure to improve this section. |
Thank you for this comment. This comment seems to be talking not about the introduction, but about section 2 (Possible mechanisms of DNA damage by UV radiation). We have expanded and reorganized this section in accordance with this remark. |
|
I suggest the following running title for the section 3: “DNA repair systems involved in the removal of photodamage” |
This has been corrected in the revised manuscript. |
|
Probably some typos (cyrillic fonts?) are reported in the figure 3 |
Thank you for this comment. This has been corrected in the revised manuscript. |
|
Section 4 is too long, is difficult for the reader to assimilate all the information. I suggest to split it into two sections: one relative to the members of the protein family and one relative to the reaction mechanisms |
Thank you for this comment. This has been corrected in the revised manuscript. |
|
Section 7 on my opinion is completely unrelated to the notions explained before. If the evolution concept is considered relevant it could be summarized in few lines and added in the introduction section. |
We believe that consideration of evolutionary issues is a built-in part of this review. Because it is impossible to fully understand the essence of biological processes and phenomena without evolutionary consideration. Understanding when and under what conditions certain proteins arose can provide additional information about the functions they perform and about possible methods of influencing them. In this regard, we left the evolution section, but made some edits to improve the overall quality of the review. |
We sincerely thank the reviewer for a thorough analysis of our work and for valuable comments that allowed us to change our review for the better.
ment.
Round 2
Reviewer 1 Report
The manuscript is now much more organized and comprehensible than before.
However, it could be much more improved and summarized.
The schemes are well but numerous.
For example, you cannot start the conclusions by a question….
See my suggestions for Abstract and for Conclusions, and please clean the rest of the manuscript.
Abstract:
Prolonged exposure to ultraviolet radiation on human skin can lead to mutations in DNA, photoaging, suppression of the immune system and other damage up to skin cancer (melanoma, basal cell and squamous cell carcinoma). We reviewed the state of knowledge of the damaging action of UVB and UVA on DNA, and also the mechanisms of DNA repair with the participation of the DNA-photolyase enzyme or of the nucleotide excision repair (NER) system. In the course of evolution, most mammals lost the possibility of DNA photo-reparation due to the damage of DNA photolyase genes, but they retained closely related cryptochromes that regulate the transcription of the NER system enzymes. We analyze the published relationships between DNA photolyases/cryptochromes and carcinogenesis, as well as their possible role in the prevention and treatment of diseases caused by UV radiation.
Conclusions:
In the actual conditions of increased in the influence of UV light and the deteriorating environmental, is not clear that the use of photolyases preventing photoaging of the skin, could avoid or treat skin diseases caused by UV damage to DNA. Sunscreens containing photolyase have shown high effectiveness, but they have not been widely used, apparently due to the relatively high cost and lack of stability of the protein extracts. Improving the properties of such creams and a wider distribution should increase their preventive role. It is possible that the use of genetic engineering methods and creation of photolyase mRNA can help people suffering from severe forms of diseases caused by the disruption of the NER system, such as xeroderma pigmentosum. The humanity is actively preparing to explore the outer space, where the influence of various types of radiation on DNA will be intense. It should be important to take a closer look at the possibility of rapid DNA photo-reparation with DNA photolyases. In addition, the accurate study of the influence of circadian rhythms and the role of cryptochromes in the regulation of cell functions, prevention and treatment of diseases caused by UV radiation, should be important.
Author Response
Replies to Reviewer 1 Comments (Round 2).
The authors are grateful to the reviewer for a careful analysis of the manuscript: biomedicines-1383821 “UV radiation in DNA damage and repair involving DNA-photolyases and cryptochromes”. We have corrected the manuscript in accordance with the obtained comments. Changes in the text were marked using the “Track changes” function of MS Word and commented below.
Comments:
The manuscript is now much more organized and comprehensible than before. However, it could be much more improved and summarized.
We sincerely thank the reviewer for a thorough analysis of our work and for valuable comments that allowed us to change our review for the better.
The schemes are well but numerous.
We agree with this comment, so Figure 4 has been removed from the text. Also we have reduced some details in the reaction mechanism of DNA-photolyase.
For example, you cannot start the conclusions by a question….
This has been corrected in the revised manuscript.
See my suggestions for Abstract and for Conclusions, and please clean the rest of the manuscript.
Thank you for this comment. We have redone the abstract and conclusion in accordance with the proposed changes. Also, some changes were made in the rest of the manuscript.

Reviewer 2 Report
Authors fulfilled all the requests, substantial changes have been made and the review has been very well re-organized.
Author Response
Replies to Reviewer 2 Comments (Round 2).
Authors fulfilled all the requests, substantial changes have been made and the review has been very well re-organized.
We sincerely thank the reviewer for appreciating our work and recommending it for publication.
Round 3
Reviewer 1 Report
I think your work is good for publication in the present form.